# Estimation of DNA Degradation in Archaeological Human Remains

**DOI:** 10.3390/genes14061238

**Published:** 2023-06-09

**Authors:** Antonella Bonfigli, Patrizia Cesare, Anna Rita Volpe, Sabrina Colafarina, Alfonso Forgione, Massimo Aloisi, Osvaldo Zarivi, Anna Maria Giuseppina Poma

**Affiliations:** 1Department of Life, Health and Environmental Sciences, University of L’Aquila, 67100 L’Aquila, Italy; antonella.bonfigli@univaq.it (A.B.); patrizia.cesare@univaq.it (P.C.); annarita.volpe@univaq.it (A.R.V.); sabrina.colafarina@univaq.it (S.C.); massimo.aloisi@studenti.unite.it (M.A.); annamariagiuseppina.poma@univaq.it (A.M.G.P.); 2Department of Human Studies, University of L’Aquila, 67100 L’Aquila, Italy; alfonso.forgione@univaq.it

**Keywords:** DNA degradation, bone remains, ancient DNA, real-time qPCR, mitochondrial DNA, nuclear DNA

## Abstract

The evaluation of the integrity and quantity of DNA extracted from archaeological human remains is a fundamental step before using the latest generation sequencing techniques in the study of evolutionary processes. Ancient DNA is highly fragmented and chemically modified; therefore, the present study aims to identify indices that can allow the identification of potentially amplifiable and sequenceable DNA samples, reducing failures and research costs. Ancient DNA was extracted from five human bone remains from the archaeological site of Amiternum L’Aquila, Italy dating back to the 9th–12th century and was compared with standard DNA fragmented by sonication. Given the different degradation kinetics of mitochondrial DNA compared to nuclear DNA, the mitochondrially encoded 12s RNA and 18s ribosomal RNA genes were taken into consideration; fragments of various sizes were amplified in qPCR and the size distribution was thoroughly investigated. DNA damage degree was evaluated by calculating damage frequency (λ) and the ratio between the amount of the different fragments and that of the smallest fragment (Q). The results demonstrate that both indices were found to be suitable for identifying, among the samples tested, those less damaged and suitable for post-extraction analysis; mitochondrial DNA is more damaged than nuclear, in fact, amplicons up to 152 bp and 253 bp, respectively are obtained.

## 1. Introduction

The analysis of DNA extracted from archeological human remains is increasingly used to investigate evolutionary processes, to reconstruct evolution models, and to study population genetics and palaeoecological changes [1,2,3]. These studies involve the latest generation sequencing techniques, the success of which is linked to the quantity and quality of the DNA extracted. Unfortunately, ancient DNA (aDNA) extracted from bone remains is scarce, highly degraded, and highly susceptible to exogenous contamination which can affect the reliability of aDNA studies [4,5]. Contamination of the aDNA can be avoided both by following the minimum guidelines during the excavation and collection of the samples and during the subsequent manipulation; aDNA should be manipulated in specialized structures and with rigorous protocols [6,7]. Studies aimed at identifying the degree of DNA degradation involve numerous research fields: cytotoxicity, cancer, forensic analysis, and aDNA studies.

When a cell dies, the endogenous endonuclease activity and spontaneous depurination cause the DNA strand to break and depending on the environmental conditions, fragments of different sizes are obtained and oxidative damage accumulates [8,9,10,11]. A number of techniques have been developed to identify and measure DNA fragmentation. A qualitative estimate of the size of the DNA fragments that have been produced can be obtained by gel electrophoresis followed by the visualization of the fragments, which is a simple approach but with limited sensitivity, Capillary electrophoresis is more sensitive but, in any case, these techniques are only able to evaluate the fragmentation and not other damage that the DNA may have suffered [12,13].

The evolution of molecular genetic techniques has allowed the amplification in PCR and qPCR and also of degraded DNA; double-strand breaks and many other forms of DNA damage block the extension step of PCR, so as the damage increases and the size of amplifiable fragments will decrease; by determining the maximum amplifiable fragment size in different samples, a relative quantification of DNA degradation can be obtained [10,14,15,16,17]. Considering degradation as a random event, in damaged samples, the amount of amplifiable fragments will decrease exponentially with increasing amplicon size, and the frequency of DNA damage (λ) can be estimated by determining the rate of decline [18]. Furthermore, it is possible to evaluate the quality of the DNA through the quantitative ratios of fragments of different sizes [19].

Until now, nearly all genetic studies performed on ancient remains have targeted mitochondrial DNA (mtDNA) genes. The mitochondrial genome is present in multiple copies in the cell [20] and is inherited from the mother; consequently, mutations are transmitted clonally across generations and can be used to trace maternal lineages [21,22,23,24,25,26,27].

However, recent studies have shown that nuclear DNA seems less subject to degradation and damage over time, probably because nuclear DNA (nucDNA) is better protected by proteins so it is possible to amplify longer strands [13,28].

Our study aims to establish an index of aDNA damage useful for identifying samples that can potentially be amplified and sequenced, reducing failures and research costs. We considered a human mitochondrial gene, mitochondrially encoded 12s RNA (12s rRNA), and a human nuclear gene, 18s ribosomal RNA (18s rRNA), in order to evaluate the extent of the damage in the two DNAs. The design of the experiments carried out to study the degradation involved the construction of pairs of primers that would allow the amplification of fragments of increasing size on the same gene and similar in size between the two genes. The set of primers allowed us to amplify in qPCR the aDNA extracted from human remains found in the cathedral of Santa Maria in Civitate in Amiternum, L’Aquila, Italy. Furthermore, the same set was used on a fragmented DNA in vitro to evaluate the differences between the two DNAs [29].

The quality of the samples will be evaluated both by determining the frequency of DNA damage (λ) and the relative ratio between the quantity of the various fragments as a function of the quantity of the smallest fragment (Q) [18,19].

Quantification of human aDNA in archaeological specimens is one of the most important factors underlying the efficiency and success of PCR-based genotyping. The ALU elements are ideal sequences for detecting small amounts of DNA due to their species specificity, small size, and extraordinarily high copy number, in this regard, we have used these sequences to quantify our samples [30].

## 2. Materials and Methods

### 2.1. Recovery of Ancient Samples

Human bone remains, dating back to the 8th–12th century AD, from which aDNA was extracted, were found in the cathedral of Santa Maria in Civitate in Amiternum, L’Aquila, Italy, during archaeological expeditions [31,32].

The samples were recovered by researchers from the Archeology Laboratory in the presence of a molecular biologist from the Genetics and Mutagenesis Laboratory of the Departments of “Human Sciences” and “Life, Health and Environmental Sciences”, respectively of the University of L’Aquila. The excavation and recovery of the archaeological remains were carried out following a protocol to prevent possible contamination of the bone remains with modern DNA [33]. The samples selected for DNA extraction were treated as in previous works [7,34]. The bone remains come from three intact graves (S.46, S.48, and S.50) and from two reworked but well-preserved graves (S.44 and S.45). The standard methods applied in physical anthropology have made it possible to determine the burial period, age of individuals at the time of death, and gender (Table 1) [35].

### 2.2. Sampling and Ethics Statement

The bone remains used were collected by Profs. Fabio Redi and Alfonso Forgione, who were responsible for the excavations on the archaeological site of Amiternum, and are stored in the Archaeology Laboratory of the Department of Human Studies, University of L’Aquila, L’Aquila, Italy. Under Italian law and in compliance with current regulations, no permits were required for the study described.

### 2.3. DNA Extraction from Bone Findings

The extraction of aDNA was carried out following the guidelines for the analysis of human skeletal remains to prevent environmental contamination. The experiments were conducted in two separate laboratories, one pre-PCR and one post-PCR, equipped with dedicated equipment [6,7]. The bone powder samples have been prepared in the pre-PCR laboratory by milling a small area of bone findings [34]. The DNA was extracted from the samples through the Geneall^®^ Exgene^TM^ Genomic DNA micro^®^ kit (Seoul, Republic of Korea) following the manufacturer’s instructions. Three DNA extractions were performed for each sample and the concentration of each sample was calculated using the standard Alu curves [30].

### 2.4. Primers: Design and Expected Amplicons

The primers were designed with Primer Express 3.0 software (Applied Biosystems, Foster City, CA, USA) using the human mitochondrial genome sequence (GeneBank: NC_012920.1) for the 12s rRNA gene and the 18s rRNA gene sequence (GenBank: NR_003286.2) in the nuclear genome; on each sequence, we built a single forward primer and several reverse primers which allowed us to obtain 5 fragments on the 18s rRNA and 12s rRNA genes of dimensions, respectively: 61, 118, 158, 253, and 332 bp and 59, 95, 152, 219, and 281 bp (Table 2, Figure 1). Pairs of primers were also constructed on the selected sequences to obtain amplicons of 2119 bp and 802 bp which contain the 12s and 18s sequences, respectively and which allowed us to quantify the amplicons obtained with the other pairs of primers (Table 2). To quantify the aDNA, the ALU sequences were amplified with a primer pair ALU50 designed to obtain a shorter amplicon.

### 2.5. Standard DNA Fragmentation

Human male genomic DNA from Jurkat cells (standard DNA) dissolved in TE buffer at a concentration of 100 ng/µL (Thermo Scientific SD1111) (Waltham, MA, USA) was fragmented by sonication with a Vibra-Cell™ Ultrasonic Sonicator (VCX 400 Sonics) (Newtown, CT, USA) with tapered microtips a diameter of 3 mm, previously washed with ethanol, at a frequency of 20 kHz, and a power of 30% with an alternating on/off pulse cycle of 8 s, and the sample is kept on ice. During the treatment, samples were taken from the DNA solution at various sonication cycles: 8, 16, 32, 64, 128, 256, 384, 996, 1252, 1508, and 1892; fragmentation was verified by electrophoresis using appropriate DNA markers. The gel was analyzed with TotalLab 1D v14.1 software, demonstrating a random fragmentation which increases with an increasing number of cycles. Appendix A shows the distribution of fragments at different sonication cycles as a function of their size; for each sample, the distribution of the fragments and the average size in base pairs with respect to the sonication cycles are reported: 1240 bp (8 cycles), 850 bp (16 cycles), 550 bp (32 cycles), 480 bp (64 cycles), 350 bp (128 cycles), 265 bp (256 cycles), 250 bp (384 cycles), 196 bp (996 cycles), 180 bp (1252 cycles), 156 bp (1508 cycles), and 150 bp (1892 cycles). DNA concentration in the different samples was measured with the Qubit™ 4 fluorometer instrument (Thermo Fisher’s) (Waltham, MA, USA).

Intact and fragmented standard DNA samples with mean sizes of 265, 196, and 156 bp were amplified by qPCR with all primer pairs of the 18s rRNA and 12s rRNA genes and used as a standard to evaluate fragmentation in aDNA samples.

### 2.6. Standard Curves and Primer Efficiency

The DNA extraction from archeological samples is fragmented and in limited quantities, therefore, its quantification is one of the most important factors at the basis of subsequent analysis. Although the quantification of fragmented DNA by qPCR has limitations [36], in our work, the qPCR amplification of small (50 bp) fragments on Alu elements, due to their extraordinarily high copy number and species specificity, has been shown to be adequate for detecting small amounts of DNA.

Standard DNA was used to construct standard curves to quantify the DNA samples extracted from bone remains, each sample was analyzed in triplicate. The ALU-50 standard curve, used to quantify the aDNA samples extracted from bone remains, was obtained by qPCR, amplifying 1 µL of template DNA from 100,000 pg/µL to 10 pg/µL, with 1:10 serial dilutions and the following pair of primers: F_ALU_50/R_ALU_50 and plotting the Ct value as a function of the Log of the concentration expressed in pg/µL (Appendix A). Furthermore, standard DNA was amplified in PCR with primers F_18s-802/R_18s-802 and F_12s-1017/R_16S-61 and amplicons of 802 and 2119 bp, respectively were obtained (Table 2). The amplicons were purified with the NucleoSpin^®^ Extract II Kits (MACHEREY-NAGEL, Düren, Nordrhein-Westfalen, Germany) following the manufacturer’s instructions, were accurately quantified with Qubit 4 Fluorometric (Thermo Fisher Scientific), and the number of copies/µL was determined through the free platform www.thermofisher.com (accessed on 12 may 2021). The standard curves as a function of concentration, expressed as number of copies/µL, were obtained from the amplification in qPCR of 1 µL of the two amplicons produced in PCR from 20,000,000 to 20 copies/µL, with serial dilutions 1:10, with the pairs of primers built on the 18s rRNA (Appendix A), and 12s rRNA (Appendix A) genes obtaining fragments of 61, 118, 158, 253, and 332 bp and 59, 95, 152, 219, and 281 bp, respectively. The curves show the Ct value as a function of the logarithm of the concentration, expressed in number of copies/µL. The straight lines obtained show a correlation coefficient R2 ≥ 0.99. The efficiency was calculated for each line, the values are between 95% and 103% demonstrating that the reaction system is valid for quantitative measurements (Table 2). The equations of the straight lines allow us to calculate the number of copies obtained by amplifying, with the different pairs of primers, both the standard DNA at different sonication times and the aDNA samples.

### 2.7. Quantitative Real-Time PCR and PCR

qPCR was performed with an ABI 7300 PCI instrument (Applied Biosystem, Foster City, CA, USA) in 96-well plates, using SYBR Green. The reaction mix (20 μL total volume) contained: 10 μL PowerUp SYBR Green Master Mix (2U×) (Applied Biosystems), 2 μL mix of specific forward and reverse primer (5 μM), DNA at various concentrations, and variable ddH_2_O up to 20 µL. Amplification conditions are as in previous works [34]. For each reaction, the efficiency (>90%), linearity, dynamic range, sensitivity, and specificity were calculated by studying the melting profiles of the amplicons. Negative controls are always present in each amplification (no DNA in the system). The relative level of inhibitors present in the extracts shifts in the Ct value for the internal PCR control (IPC) in the real-time PCR assay were checked. All samples were amplified in triplicate.

PCR was conducted with the Hybaid PCR Express ThermoCycler instrument using the KAPA2G Fast HotStart ReadyMix 2x kit (Kbiosystems, Cape Town, South Africa) following the manufacturer’s instructions

### 2.8. Quantification of aDNA Fragmentation

The degree of DNA fragmentation was evaluated by calculating two different indices, λ and Q. The concentrations of the various amplified fragments (amplicons) in the different DNA samples, expressed as a number of copies/μL, were calculated using the standard curves. The frequency of DNA damage (λ) can be estimated by determining the decline rate of amplification of fragments of increasing size. According to a random degradation model, the amount of available templates will decrease exponentially as the fragment size increases in the damaged samples. By plotting the logarithm of the number of copies obtained for each fragment size as a function of the size of the fragments, we obtain a straight line whose slope λ represents the probability of a nucleotide being damaged. The straight line is described by the equation:$$Log\left(A_{x}\right)=Log\left(N\right)-\lambda x$$
where *A_x_* is the copy number corresponding to each fragment sizes (*x*), *N* is the maximum number of amplifiable fragments, and *λ* is the probability of a nucleotide being damaged (*λ*) [18].

Since DNA fragmentation has a greater impact on the amplification of longer targets, the degree of fragmentation of a DNA sample can also be inferred by calculating the fraction of DNA (*Q*) amplified with larger *amplicons* (*x*) relative to the *smaller amplicon* (61 bp and 59 bp for 18s rRNA and 12s rRNA, respectively):$$Q=\frac{n{}^{\circ}\;copies\;of\;amplicon\;x}{n{}^{\circ}\;copies\;of\;smaller\;amplicon}.$$

*Q* will have a value between 0 and 1 and can be used as a relative measure of DNA quality [19].

### 2.9. Electrophoresis

Both sonicated samples and qPCR products were subjected to an agarose gel electrophoretic run (0.8% to 1.8%, depending on the size of the fragments); the DNA ladder is always loaded as PM marker (SHARPMASS™ 50- Ready to load DNA ladder, Euroclone S.p.A.; 100 bp DNA ladder Invitrogen; 1 kb DNA ladder Sigma Aldrich). The electrophoretic run was carried out at constant 80 Volts in 40 mM Tris-acetate-EDTA buffer. The gels were stained with 0.5 μg/mL ethidium bromide; the bands obtained were acquired and analyzed with the UVITEC Alliance Q9 system (Uvitec Ltd., Cambridge, UK) and the TotalLab 1D v14.1 software (TotalLab, Newcastle upon Tyne, UK).

## 3. Results

### 3.1. Standard DNA Fragmentation Analysis Subsection

Human male genomic DNA from Jurkat cells dissolved in TE buffer at a concentration of 100 ng/µL (Thermo Scientific SD1111) (standard DNA) was sonicated as described in Materials and Methods; sonicated and unsonicated DNA (control DNA, Ctr) was subjected to qPCR with all primer pairs built on the 18s rRNA and 12s rRNA gene, in order to estimate whether the proposed method is capable of assessing aDNA fragmentation.

Among the samples through sonication (Appendix A), those at 256 (DNA 256), 996 (DNA 996), and 1508 (DNA 1508) cycles were chosen, these had mean fragment sizes of 265 bp, 196 bp, and 156 bp, respectively (Figure 2). 

The choice of the three sonicated samples takes into account that the aDNA is highly fragmented with a size range of 50–300 bp [4,37].

Figure 3 shows the thermal denaturation profiles of the qPCR performed on the control DNA with all the constructed primers and the subsequent electrophoretic run of the products obtained. The analysis of the thermal denaturation profiles highlights a single peak demonstrating the specificity of the reaction and the absence of non-specific products, these data are confirmed by the presence of a single band in the subsequent electrophoretic run. The electrophoretic run confirms the expected amplicon sizes with the various primer pairs on the 18s rRNA 12s rRNA genes as reported in materials and methods (Figure 1); in fact, amplicons, respectively 61,118, 158, 253, and 332 bp and 59, 95, 152, 219, and 281 bp in sizes are obtained.

Figure 4 shows representative amplification curves of the 18s rRNA (Figure 4A) and 12s rRNA (Figure 4C) genes with threshold cycles (Ct) trends, as a function of the amplicon size for all the DNA samples analyzed (Figure 4B,D).

The amplification of fragments of different sizes shows overlapping curves in control DNA only, indicating that amplification is independent of fragment size when the DNA is unbroken. In sonicated samples, the curves shift to the right with increasing fragment size (>Ct). This aspect is better highlighted in the histograms (Figure 4B,D) where we see that as the size of the amplicon increases, the differences in Ct between the control DNA and those differently fragmented increase; in particular, in the amplification of the 332 bp and 281 bp fragments, for 18s and 12s, respectively, the Ct of DNA 1508 increase by 10 cycles compared to the control DNA.

### 3.2. Evaluation of Fragmentation in Standard DNA Subsubsection

Ct obtained from qPCR amplification, with the different pairs of primers, of sonicated and unsonicated standard DNA, through the standard curves (Appendix A) constructed as described in Materials and Methods, allowing for calculating concentration expressed as the number of copies for each sample. To evaluate the quality of the sonicated or unsonicated standard DNA, two indices λ and Q were calculated, the equations of which are reported in Materials and Methods.

λ represents the frequency of DNA damage and is obtained by determining the rate of decline of amplification with increasing amplicon size.

Figure 5 shows the logarithm of the number of copies as a function of the size of the various amplicons for each DNA sample and for the two genes considered; the slope of the lines obtained by linear interpolation of the experimental values represents λ. Both nucDNA (Figure 5A) and mtDNA (Figure 5B) in the unsonicated sample are unbroken, as λ has very low values (0.0002 and 0.0001, respectively). In DNA 256 and DNA 996, intermediate degree of fragmentation, nucDNA has a smaller λ (0.0024 and 0.0045) than mtDNA (0.005 and 0.008), indicating a greater lability of the latter; in DNA 1508, maximum fragmentation, the rate of decline is comparable for nuclear (0.0101) and mitochondrial (0.0112) DNA.

Q score represents the fraction of DNA amplified with amplicons of different sizes compared to the smallest amplicon: 61 bp and 59 bp for 18s rRNA and 12s rRNA, respectively (n° copies of amplicon x/n° copies of smaller amplicon); it lies between 0 and 1 where 1 indicates the highest degree of integrity.

Figure 6 shows the Q values, calculated for all fragments, obtained from the amplification of the 18s rRNA (Figure 6A) and 12s rRNA (Figure 6B) genes on all DNA samples. Considering the control DNA, we see that Q stands at values ranging from 0.75 to 0.9 which indicates little-fragmented DNA. Considering the DNAs 256 and 996, we see a significant variation for the nuclear gene with respect to Q118/61 starting from Q253/61, while for the mitochondrial gene, already from Q152/59, there is a very significant decrease with respect to Q95/59. Thus, showing that at intermediate sonication cycles the nuclear DNA is of better quality than the mitochondrial one. Taking into consideration DNA1508, subjected to the highest cycles of sonication, it is evident that both mitochondrial and nuclear DNA are strongly compromised as Q values are extremely low.

### 3.3. Evaluation of Fragmentation in aDNA

The 18s rRNA and 12s rRNA genes were amplified in qPCR with all pairs of primers (Table 1 and Figure 1) on the DNA samples extracted from archaeological finds S44, S45, S46, S48, and S50 to evaluate aDNA quality.

Figure 7, panel A and panel D, show the dissociation curves of the fragments obtained from aDNA amplification on both the 18s rRNA gene and 12s rRNA, respectively belonging to a representative sample. It is evident that the aDNA does not show amplification with the primers from which the 332 bp fragment on the 18s rRNA gene is obtained and with the primers from which the 219 and 281 bp fragments on the 12s rRNA gene are obtained. These results are confirmed by the subsequent electrophoretic run (Figure 7B,E) in which the aDNA amplicons were compared with those obtained on the standard control DNA (lines 3–11); from the latter, all the expected amplicons are obtained (lines 1,12). This provides initial qualitative evidence of aDNA fragmentation and, at the same time, the lack of amplicons above 300 bp demonstrates the absence of contamination by modern DNA. The fragments obtained from the amplification, with all pairs of primers, of the 18s rRNA (Figure 7C) and 12s rRNA (Figure 7F) genes, both on the aDNA extracted from all the bone remains considered and on the standard control DNA, were mixed and subjected to an electrophoretic run. Results confirm that on all the aDNA samples, the amplification of the 332 bp fragment on the nuclear DNA and of fragments 219 on and 281 bp on the mitochondrial DNA is not obtained (lines 2–6), while on standard DNA 1508 all the expected amplicons are obtained (line 1).

The Ct obtained from qPCR amplification with the different pairs of primers, of all aDNA samples, through the standard curves (Appendix A) constructed as described in Materials and Methods, allow for calculating the concentration expressed as a number of copies for each sample. To evaluate the quality of the aDNA, two indices, λ and Q, were calculated the equations of which are reported in Materials and Methods.

Figure 8 shows the logarithm of the number of copies as a function of the size of the various amplicons for each aDNA sample and for the two genes considered; the slope of the lines obtained by linear interpolation of the experimental values represents λ.

The Q values, calculated for all fragments obtained from the amplification of the 18s rRNA and 12s rRNA genes on all aDNA samples, are shown in Figure 9.

The aDNA from bone remains S.45 and S.48 are the most intact as regards the amplification of the 18s rRNA gene with a λ of 0.043 and 0.040, respectively while S.44 and S.50 appear to be those with a higher rate of decline with λ 0.0078 and 0.0080, respectively (Figure 8A). Taking into consideration the Q parameter, S.45 and S.48 confirm the good quality of their nucDNA with a Q253/61 of 0.70 and 0.71, respectively, therefore, 70% of the fragments have dimensions up to 253 bp; S.44 turns out to be the worst with less than 1.7% of fragments being up to 253 bp in size; S.46 and S.50 are comparable to each other, less than 40% of the fragments are larger than 253 bp (Figure 9A).

Considering the λ parameter calculated on the mtDNA, samples S.45 and S.46 are the best with values of 0.0048 and 0.0070, respectively (Figure 8B), a result confirmed by the parameter Q152/59, equal to 0.75 and 0.60, respectively, according to which more than 60% of the fragments are larger than 152 bp (Figure 9B).

S.44 and S.50 are the bone remains with the most damaged mtDNA; these have λ values of 0.0144 and 0.0134, respectively (Figure 8B) and less than 20% of the fragments with dimensions greater than 152 bp (Q152/59 0.19 and 0.05, respectively) (Figure 9B).

The results obtained show that mtDNA is more damaged than nucDNA, in fact, mitochondrial aDNA amplifies up to 152 bp while nuclear aDNA amplifies up to 253 bp in all bone remains; moreover, in mtDNA, the 152 bp amplicon is present with percentages ranging from 75 to 5%; while in nucDNA, the 158 bp amplicon is present with percentages ranging from 93 to 77%.

## 4. Discussion and Conclusions

Studies on ancient DNA (aDNA) present critical issues related to low quantities obtained in the extraction processes, degradation of the samples, and high risk of contamination [4]. The methodological improvements have given impetus to research and have made it possible to deepen evolutionary processes, to reconstruct evolution models, and to study population genetics and palaeoecological changes [1,2,3]. These studies involve Next-Generation Sequencing techniques, the success of which, however, is linked to the evaluation of the quantity and quality of the extracted DNA.

Studies on damaged DNA extracted from different types of samples (forensic, anthropological, etc.) have taken into consideration the possibility of successfully amplifying both nuclear and mitochondrial genes; however, studies on aDNA have mainly focused on the analysis of mtDNA. The choice of mtDNA derives both from its higher copy number in the cell and from the possibility of tracing maternal lineages since, being mtDNA is of maternal inheritance, mutations are transmitted clonally through the generations. In recent times, the different vulnerability to nuclear and mitochondrial DNA damage has also been taken into consideration [13,28,38].

The purpose of this work was to identify DNA damage indices useful for individuate samples that potentially can be amplified and sequenced, thus reducing failures and costs in the research. A mitochondrial gene, mitochondrially encoded 12s RNA (12s rRNA), and a nuclear gene, 18s ribosomal RNA (18s rRNA), were taken into consideration in order to evaluate the extent of the damage in the two DNAs.

The design of primers that would allow for the amplification of fragments of increasing size on the same gene and similar in size between the two genes and makes it possible to consider two indices: λ, which represents the frequency of DNA damage, obtained by determining the rate of decline of amplification with increasing amplicon size [18], and Q, which represents the fraction of DNA amplified with amplicons of different sizes relative to the smaller amplicon [19]. These indices allowed us to evaluate the quality of the extracted aDNA. λ and Q were calculated on fragmented DNA in vitro with increasing cycles of sonication to simulate the wide range of fragment sizes obtained post-mortem [29].

The results obtained on standard sonicated DNA confirm evidence reported in the literature regarding the greater vulnerability to damage of mitochondrial DNA when placed in the same conditions (acellular systems) as nuclear DNA [28]; in fact, the λ parameters are all lower in nuclear DNA than in mitochondrial DNA, and the decrease in the Q index is always very significant in mitochondrial DNA, while in nuclear, it only starts at Q253/6.

However, the Q score (Figure 6) decreases with increasing sonication cycles and amplicon size in both 18s rRNA and 12s rRNA; this result confirms data reported in the literature regarding the analysis of physical fragmentation [29]. In fact, the Q score trend is consistent with the distribution of the dimensions of the fragments obtained from the analysis of the electrophoretic run of sonicated and unsonicated DNA (Figure 2). Thus, the parameters Q and λ are more suitable for assessing the fragmentation of standard DNA.

Damage to aDNA presents characteristics that are different from the physical fragmentation obtained with standard DNA. In fact, analyzing the Q118/61 and Q158/61 ratios obtained by amplifying the 18srRNA gene on the aDNA, we find values greater than or equal to 0.8 in all the aDNA samples extracted from the bone remains considered, therefore, comparable to the corresponding Q value in the unsonicated standard DNA. However, in the aDNA, amplicons above 253 bp are not obtained, while in the DNA Standard at the maximum degree of sonication (DNA 1508), amplicons up to 332 bp are obtained.

The different aDNA samples extracted from bone remains differ starting from Q253/61; this parameter decreases very significantly while remaining above 0.6, in S45 and S48, while in the DNA256 sample (lower degree of sonication), it has a value of 0.4. As a result, S45 and S48 appear to be reliable samples as they amplify fragments of 253 bp with a good percentage, a fundamental characteristic in generally low-yielding aDNA extractions.

In mtDNA, no amplification is observed on the 12s rRNA gene above 152 bp but when analyzing the values of Q95/59 in S.45, S.46, and S.48, they are greater than or equal to 0.8, therefore, comparable to the corresponding Q value in unsonicated standard DNA.

The same aDNA samples have a Q152/59 which decreases significantly but with values higher than 0.33 value of the DNA 256 sample (lower degree of sonication). Paradoxically, however, we find that in standard DNA at maximum sonication (DNA1508), amplicons reaching up to 281 bp are obtained.

In this case as well, the Q parameter allows for identifying the finds that are most suitable for use in genetic analyses following extraction. Furthermore, this parameter allows us to exclude the S44 and S50 samples in studies on both mtDNA and nucDNA.

The λ parameter also allows for the identification of the less damaged bone findings, and the results are in line with those obtained with the Q parameter. In fact, as regards 18srRNA, the best samples are S45 and S48 with λ values of 0.0043 and 0.0040, respectively, and in terms of 12s rRNA, the best samples are S45, S46, and S48 with λ values of 0.0048, 0.0070, and 0.0110, respectively.

It can, therefore, be stated that, even in bone samples, mtDNA shows greater vulnerability than nucDNA. It is likely that the presence of accessory proteins, it in some way protects nucDNA by reversing the trend normally observed in biological samples where compartmentalization protects mtDNA from degradation by nucleases [28,39].

As reported in literature, results show that the extent of DNA degradation is not linked either to the state of the tomb or to the age of the find. In fact, sample S.50, one of the worst in terms of DNA damage, is the most recent sample, dating back to the 12th century, and derives from an intact tomb, thus degradation is connected with the conservation conditions of the find and the environmental stresses to which it was subjected to rather than age or state of the tomb [37,38,40,41,42,43,44].

In conclusion, even if the results obtained have not been verified by sequencing tests, both indices considered in this study allow us to identify, among the available samples, those most suitable to be used in post-extraction analyses. These indices take into account not only the fragmentation of the DNA but also the further damages suffered by the genetic material which prevent its amplification and, in fact, give us indications on the dimensions of the amplifiable fragments.

These studies can also find application in the diagnostic field (fragmentation of sperm DNA, DNA released from cells) in the forensic field (degradation of forensic samples) and, since qPCR is a very sensitive analytical technique, λ and Q also allow us to discriminate between samples of very poor quality and quantity. Authors should discuss the results and how they can be interpreted from the perspective of previous studies and of the working hypotheses. The findings and their implications should be discussed in the broadest context possible. Future research directions may also be highlighted.

## Figures and Tables

**Figure 1 genes-14-01238-f001:**
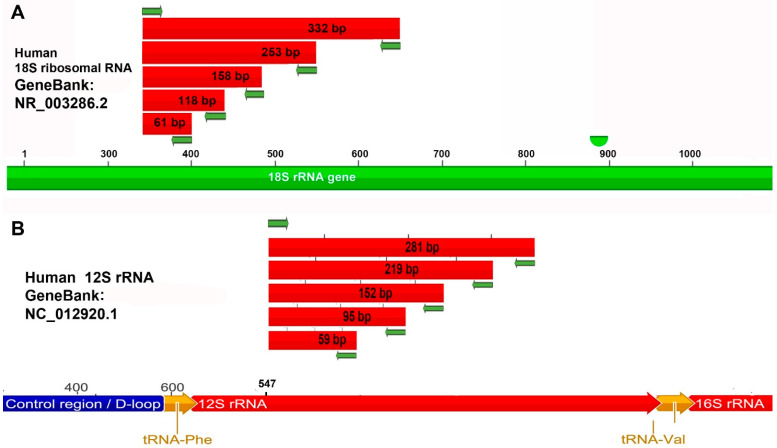
Design of primers to evaluate DNA fragmentation. The diagram shows the position of the primers on the 18s rRNA (**A**) and 12s rRNA (**B**) gene sequences and the length of the amplicons produced in qPCR.

**Figure 2 genes-14-01238-f002:**
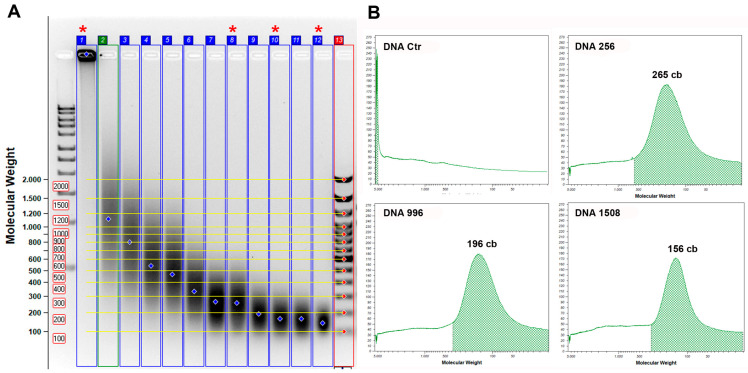
Standard DNA samples chosen for qPCR analysis. Electrophoretic analysis of human male genomic DNA samples from Jurkat cells, (SD1111 Thermo Scientific, Waltham, MA, USA) (standard DNA) sonicated and not. (**A**) Electrophoretic run of samples in 1.4% agarose gel: lane 1-1 kb Sigma Aldrich DNA ladder; lane 2—control DNA; lanes 3-13—DNA sonicated at increasing times; lane 14—10 bp Invitrogen DNA ladder; lines with asterisks (2, 9, 11, and 13) correspond to the samples chosen for qPCR analysis. (**B**) distribution of fragments and average size in base pairs of the selected samples (Ctr, DNA256, DNA996, and DNA1508).

**Figure 3 genes-14-01238-f003:**
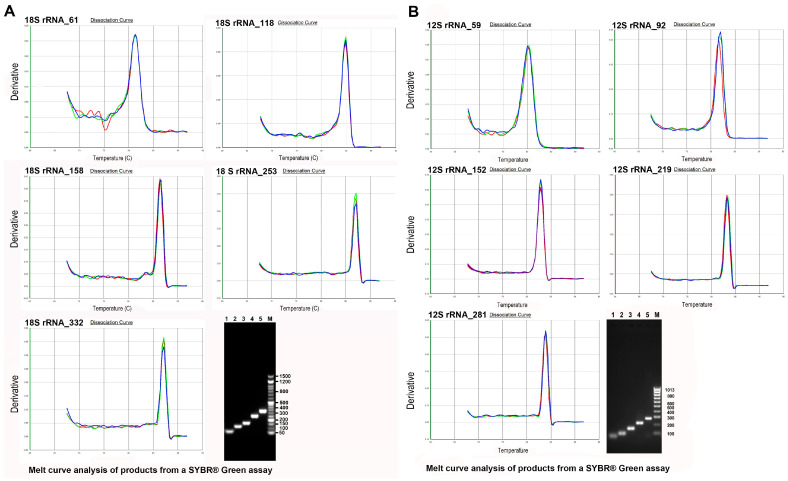
Thermal denaturation curves and electrophoresis of the amplicons produced in qPCR on the 18s rRNA and 12s rRNA genes of control DNA. Amplification products were obtained with 2 ng of unsonicated standard DNA (Ctr) in 20 µL of reaction system. (**A**) Thermal denaturation curves of the amplicons produced on the 18s rRNA gene (61, 118, 158, 253, and 332 bp) and respective electrophoretic run on 1.8% agarose gel: lanes 1, 2, 3, 4, and 5 bands 61, 118, 158, 253, and 332 bp, respectively; M marker SHARPMASS™ 50-Ready to load DNA ladder (Euroclone S.p.A.). (**B**) Thermal denaturation curves of the amplicons produced on the 12s rRNA gene (59, 95, 152, 219, and 281 bp) and respective electrophoretic run on 1.8% agarose gel: lanes 1, 2, 3, 4, and 5 bands 59, 95, 152, 219, and 281 bp, respectively; M marker SHARPMASS ™ 50—Ready to load DNA ladder (Euroclone S.p.A.).

**Figure 4 genes-14-01238-f004:**
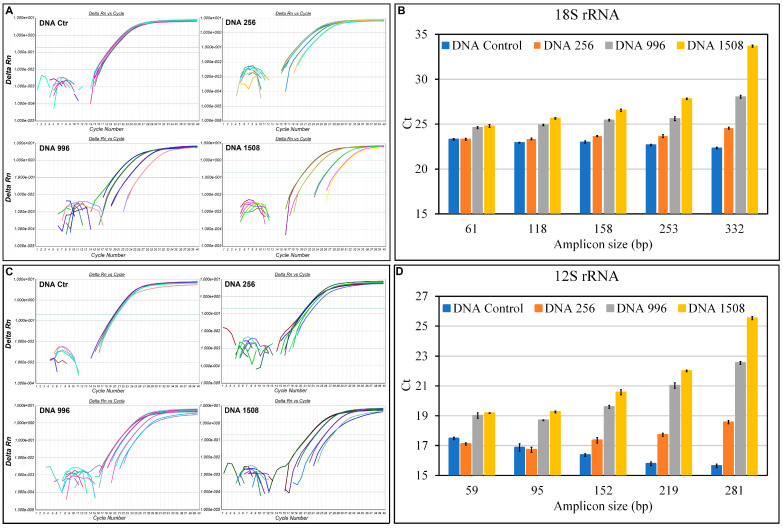
qPCR amplification of the 18s rRNA and 12s rRNA genes on sonicated and unsonicated standard DNA. The amplification products were obtained with 2 ng of standard DNA unsonicated (Ctr) and sonicated (DNA256, DNA996, and DNA1508 with mean fragment sizes of 265, 196, and 156 bp, respectively) in 20 µL of reaction system. (**A**) Amplification plot of the 18s rRNA gene with the different primer pairs on the sonicated and unsonicated standard DNA. (**B**) Ct trend as a function of the size of the amplicons produced on the 18s rRNA gene with the different primer pairs (61, 118, 158, 253, and 332 bp). Means ± SE are reported. (**C**) Amplification plot of the 12s rRNA gene with the different primer pairs on the sonicated and unsonicated standard DNA. (**D**) Ct trend as a function of the size of the amplicons produced on the 12s rRNA gene with the different primer pairs (59, 95, 152, 219, and 281 bp). Means ± SE are reported.

**Figure 5 genes-14-01238-f005:**
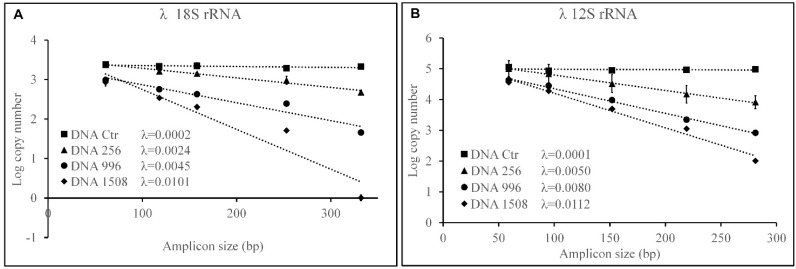
Estimation of damage to standard DNA sonicated and not λ values. Logarithm of the number of copies amplified on the 18s rRNA (**A**) and 12s rRNA (**B**) genes as a function of the amplicon size on the samples DNA unsonicated (Ctr) and sonicated (DNA 256, DNA 996, and DNA 1508 with mean fragment sizes of 265, 196, and 156 bp, respectively). The values of λ are shown for all the samples analysed, it represents the estimated probability that a nucleotide is damaged and it is obtained from the linear interpolation of the values. Values are reported as means ± SE.

**Figure 6 genes-14-01238-f006:**
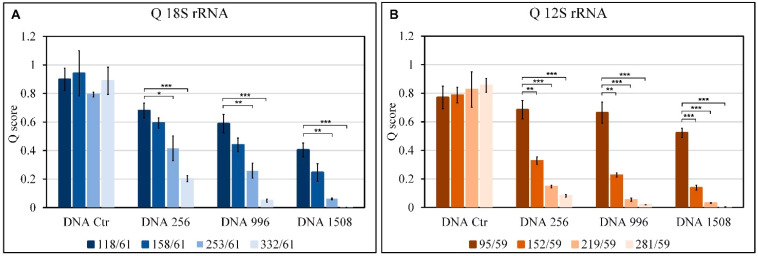
Estimation of damage to standard DNA sonicated and not Q values. Q score, calculated for all fragments, obtained from amplification of the 18s rRNA (**A**) and 12s rRNA (**B**) genes on the samples DNA unsonicated (Ctr) and sonicated (DNA 256, DNA 996, DNA 1508 with mean fragment sizes of 256, 196 and 156 bp, respectively). Q represents the fraction of DNA amplified with amplicons of different sizes compared to the smallest amplicon: 118/61, 158/61, 253/61, and 332/61 for 18s rRNA and 95/59, 152/59, 996/59, and 281/59 for 12s rRNA (see Materials and Methods). Means ± SEM are reported (n = 3). The Student’s *t*-test was employed to compare the Q values 118/61 and 95/59 for 18s rRNA and 12s rRNA, respectively with the other Q values for all samples. * *p* < 0.05; ** *p* < 0.005; *** *p* < 0.0005.

**Figure 7 genes-14-01238-f007:**
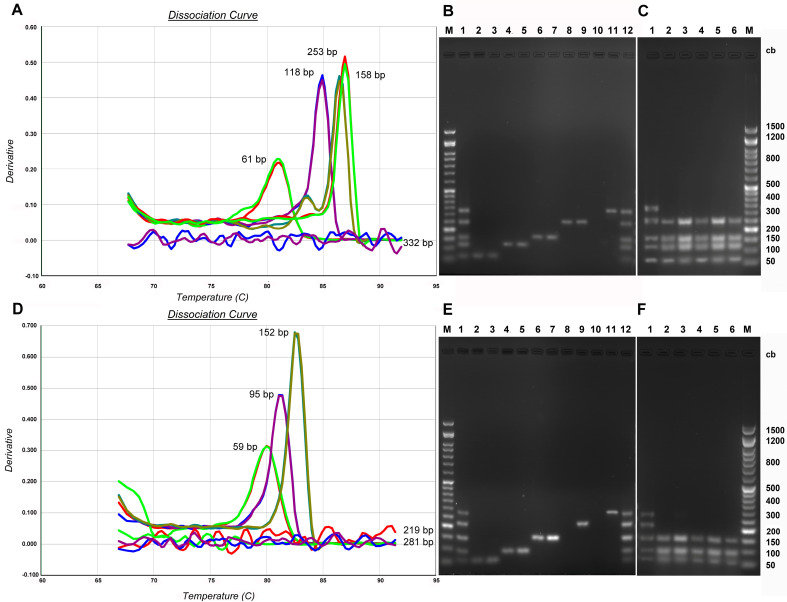
Qualitative evaluation of the aDNA. Panels (**A**–**C**) show the qPCR amplifications of the 18s rRNA gene from which fragments of 61, 118, 158, 253, and 332 bp are obtained. Panels (**D**–**F**) show the qPCR amplifications of the 12s rRNA gene from which fragments of 59, 95, 152, 219, and 281 bp are obtained. (**A**,**D**) Melting curve profiles of the amplicons produced on the 18s rRNA and 12s rRNA genes obtained in qPCR on the aDNA of the S.44 bone remnant (500 pg in system). (**B**,**E**) 2% agarose gel electrophoresis of the amplicons produced on the 18s rRNA and 12s rRNA genes; line 1, 12—mix (6 µL of each system) of the amplicons obtained in qPCR on the standard control DNA; lines 2, 4, 6, 8, and 10—amplicons obtained in qPCR on the S.44 aDNA; lines 3, 5, 7, 9, 11—amplicons obtained in qPCR using standard control DNA as templates; M marker (SHARPMASS™ 50-Ready) (**C**,**F**) 2% agarose gel electrophoresis of the amplicons produced on the 18s rRNA and 12s rRNA genes; line 1—mix (6 µL of each system) of the amplicons obtained in qPCR on DNA 1508; lines 2,3,4,5,6—mix (6 µL of each system) of the amplicons obtained in qPCR on the aDNA of bone remains S.44, S.45, S.46, S.48, and S.50, respectively; M marker (SHARPMASS™ 50-Ready).

**Figure 8 genes-14-01238-f008:**
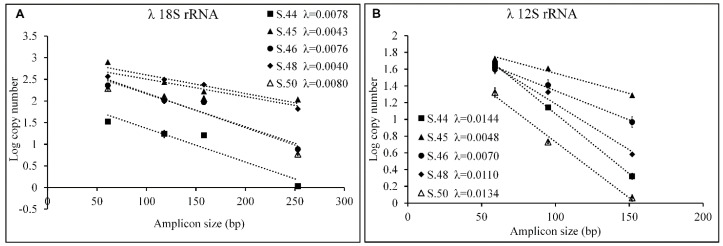
Estimation of damage to aDNA from bone remains, λ values. Logarithm of the number of copies amplified on the 18s rRNA (**A**) and 12s rRNA (**B**) genes as a function of amplicon size on the different aDNA samples (S.44, S.45, S.46, S.48, and S.50). The values of λ are reported for all the samples analyzed, it represents the estimated probability that a nucleotide is damaged and is obtained by the linear interpolation of the values. Values are reported as means ± SE.

**Figure 9 genes-14-01238-f009:**
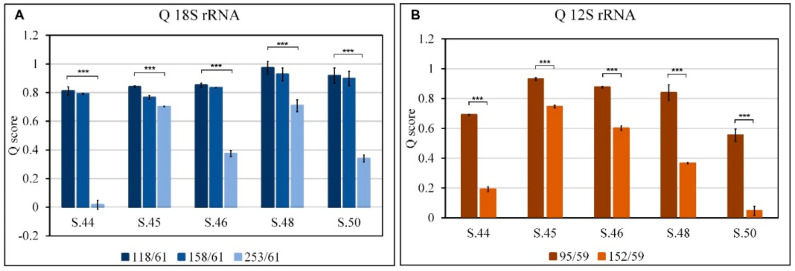
Estimation of damage to aDNA from bone remains, Q values. Q score, calculated for all fragments obtained from amplification of the 18s rRNA (**A**) and 12s rRNA (**B**) genes on the different different aDNA samples (S.44, S.45, S.46, S.48, S.50). Q represents the fraction of DNA amplified with amplicons of different sizes compared to the smallest amplicon: 118/6, 158/61, 253/61 and 332/61 for 18s rRNA and 95/59, 152/59, 996/59 and 281/59 for 12s rRNA, (see Materials and Methods). Means ± SE are reported (n = 3). The Student’s *t*-test was employed to compare the Q values 118/61 and 95/59 for 18s rRNA and 12s rRNA, respectively with the other Q values for all samples. *** *p* < 0.0005.

**Table 1 genes-14-01238-t001:** Scheletal remains analyzed.

Sample	Bone	Age (Years)	Sex	Period
S.44	Right femur	45–50 (Adult)	Female	9th–10th century
S.45	Right femur	45–50 (Adult)	Male	9th–10th century
S.46	Right femur	16–18 (Teenager)	Male	9th–10th century
S.48	Left femur	45–50 (Adult)	Male	9th century
S.50	Right femur	45–50 (Adult)	Female	12th century

**Table 2 genes-14-01238-t002:** List of primers used.

Primer	Primer Sequence 5′-3′	Tm	Product (bp)	Efficiency
F_18s-802	GGCGGCTTTGGTGACTCTA	58.8 °C	802	
R_18s-802	TGGTCGGAACTACGACGGTAT	59.8 °C		
F_12s-1017	TGTTTAGACGGGCTCACATCA	57.9 °C	2119	
R_16S-61	CTCCATAGGGTCTTCTCGTCTTG	62.4 °C		
F_ALU_50 R_ALU_50	GATCACGAGGTCAGGAGGTC GGGTTTCACCGTTTTAGCCG	61.4 °C 59.4 °C	50	98%
F_18s	CGAACGTCTGCCCTATCAACTT	60.3 °C		
R_18s-61	ACCCGTGGTCACCATGGTA	58.8 °C	61	100%
R_18s-118	GGATGTGGTAGCCGTTTCTCA	59.8 °C	118	97%
R_18s-158	GGGTCGGGAGTGGGTAATTT	59.4 °C	158	97%
R_18s-253	CCAATGGATCCTCGTTAAAGGA	58.4 °C	253	101%
R_18s-332	CGAGCTTTTTAACTGCAGCAACT	58.9 °C	332	95%
F_12s	AAAACTGCTCGCCAGAACACTAC	60.6 °C		
R_12s-59	GCACCGCCAGGTCCTTT	57.6 °C	59	100%
R_12s-95	TCGATTACAGAACAGGCTCCTCTA	61.0 °C	95	99%
R_12s-152	TGCTGAAGATGGCGGTATATAGG	60.6 °C	152	97%
R_12s-219	TGGGCTACACCTTGACCTAACG	62.1 °C	219	103%
R_12s-281	TTTCATAAGGGCTATCGTAGTTTTCTG	60.4 °C	281	100%

## Data Availability

The data presented in this study are available in the present article and on request to the corresponding authors.

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
