# Peer review of "Estimation of DNA Degradation in Archaeological Human Remains"

_genes, 2023, doi:10.3390/genes14061238_

Round 1
Reviewer 1 Report
The authors should discuss the limitations of qPCR quantitation of aDNA. See "Gill, Peter, Øyvind Bleka, and Ane Elida Fonneløp. "Limitations of qPCR to estimate DNA quantity: An RFU method to facilitate inter-laboratory comparisons for activity level, and general applicability." Forensic Science International: Genetics 61 (2022): 102777"
Author Response
Dear Reviewer,
We would like to thank you for your time and effort in revising our paper and for offering us criticisms and suggestions for improving the final manuscript. We accurately read, considered your comments, and modified our manuscript according to your suggestions, whenever possible. Please find enclosed a point-by-point answers to the comments.
[Genes] Manuscript ID: genes-2382442
Reviewer 1: The authors should discuss the limitations of qPCR quantitation of aDNA. See "Gill, Peter, Øyvind Bleka, and Ane Elida Fonneløp. "Limitations of qPCR to estimate DNA quantity: An RFU method to facilitate inter-laboratory comparisons for activity level, and general applicability." Forensic Science International: Genetics 61 (2022): 102777"
R: Thank you for your suggestion. We consider the bibliographic entry very relevant and interesting for its analysis on the limits of calculating the concentration of DNA in samples that have various degrees of degradation such as forensic and ancient DNA samples. We have acknowledged the critical points by inserting the bibliographic entry in paragraph 2.6.
However, in our work on aDNA we have used qPCR amplification of Alu elements as a sensitive and accurate marker. We were able to verify that the amplification of Alu50 provided reliable data on differently fragmented DNA.
Reviewer 2 Report
An interesting paper which convincingly backs up the finding that mtDNA is more susceptible to fragmentation that nuDNA in ancient samples, and presents a potential way of screening samples for damage prior to sequencing.
One minor note regarding the sonicated DNA simulation of damage, I found the nomenclature of the DNA used in the qPCR tests a bit confusing. I know that the sizes of sonicated DNA used vs the number of cycles is explained in the Materials and Methods section, but flicking back and forward a few times to remind myself of the different fragment sizes (eg for figure 4) was a little frustrating. It could be useful to put a note of the average DNA sizes of the different sonicated products in the caption?
The paper was in large well written and easy to understand.
A few minor typos to address:
line 217 - in the equation, change the 'o' to an 'a' in "smoller"
line 222 - in the heading, 'electroforesis' - change to 'electrophoresis'
Author Response
Dear Reviewer,
We would like to thank you for your time and effort in revising our paper and for offering us criticisms and suggestions for improving the final manuscript. We accurately read, considered your comments, and modified our manuscript according to your suggestions, whenever possible. Please find enclosed a point-by-point answers to the comments.
[Genes] Manuscript ID: genes-2382442
Reviewer 2: An interesting paper which convincingly backs up the finding that mtDNA is more susceptible to fragmentation that nuDNA in ancient samples, and presents a potential way of screening samples for damage prior to sequencing.
One minor note regarding the sonicated DNA simulation of damage, I found the nomenclature of the DNA used in the qPCR tests a bit confusing. I know that the sizes of sonicated DNA used vs the number of cycles is explained in the Materials and Methods section, but flicking back and forward a few times to remind myself of the different fragment sizes (eg for figure 4) was a little frustrating. It could be useful to put a note of the average DNA sizes of the different sonicated products in the caption?
The paper was in large well written and easy to understand. A few minor typos to address:
line 217 - in the equation, change the 'o' to an 'a' in "smoller"
line 222 - in the heading, 'electroforesis' - change to 'electrophoresis'
R: Thank you for your suggestion. We added the requested references: we have inserted in the captions of Figures 4,5,6 the average DNA sizes of the different sonicated products; line 217 in the equation, we changed the 'o' to an 'a' in "smoller"; line 222 we changed in the header, 'electroforesis' - to 'electrophoresis'.
Reviewer 3 Report
Dear Authors,
Nice and well-done work. The presentation was very good and understandable. The study presented provides a very useful database of the sector. The research regarding the DNA is very useful, because of the necessity of doing such of analysis, and also because of the material degradation in some cases.
Author Response
Dear Reviewer,
We would like to thank you for your time and effort in revising our paper and for offering us criticisms and suggestions for improving the final manuscript. We accurately read, considered your comments, and modified our manuscript according to your suggestions, whenever possible. Please find enclosed a point-by-point answers to the comments.
[Genes] Manuscript ID: genes-2382442
Reviewer 3: Dear Authors, Nice and well-done work. The presentation was very good and understandable. The study presented provides a very useful database of the sector. The research regarding the DNA is very useful, because of the necessity of doing such of analysis, and also because of the material degradation in some cases.
R: Thank you for your comments.